# Artificial Intelligence in Biomedicine: A Legal Insight

**DOI:** 10.3390/biotech10030015

**Published:** 2021-07-14

**Authors:** Takis Vidalis

**Affiliations:** Hellenic National Commission for Bioethics and Technoethics, 10674 Athens, Greece; t.vidalis@bioethics.gr

**Keywords:** artificial intelligence, biomedicine, data protection, medical duty, informed consent, unsupervised systems

## Abstract

The involvement of artificial intelligence in biomedicine promises better support for decision-making both in conventional and research medical practice. Yet two important issues emerge in relation to personal data handling, and the influence of AI on patient/doctor relationships. The development of AI algorithms presupposes extensive processing of big data in biobanks, for which procedures of compliance with data protection need to be ensured. This article addresses this problem in the framework of the EU legislation (GDPR) and explains the legal prerequisites pertinent to various categories of health data. Furthermore, the self-learning systems of AI may affect the fulfillment of medical duties, particularly if the attending physicians rely on unsupervised applications operating beyond their direct control. The article argues that the patient informed consent prerequisite plays a key role here, not only in conventional medical acts but also in clinical research procedures.

## 1. Introduction

Developments in contemporary biomedicine raise ethical and legal questions relevant to the extensive use of artificial intelligence (AI) applications both in conventional medical practice and in research activities [1] (p. 5). With no doubt, the introduction of algorithms promises better results in the evaluation of specific cases, if these algorithms are formed and constantly updated on the basis of appropriate statistical information deriving from clinical studies with similar characteristics. On the other hand, AI applications as substitutes of individual physicians, namely human decision-makers, do not always ensure the best option is followed for a particular patient, even if decisions they recommend are evidence-based [2] (p. 400). Indeed, statistical evidence does not necessarily capture the complex nature of specific clinical cases; medical practice cannot be reduced to pure mathematical models. That is why the involvement of AI systems and the extent of their use by attending physicians are topics that influence the patient/physician relationship in terms of ethics and law.

We can distinguish two central questions concerning the use of AI in biomedicine from a legal standpoint.

First, we encounter a question referring to the formation of algorithms suitable for supporting medical decision-making. This work presupposes extensive processing of massive information, including scientific information, statistical data, and personal data of health importance (genetic, clinical, and lifestyle data) [1]. The collection and processing of personal data in particular are subject to the data protection legal framework.

The second question is relevant to the influence of the AI automated decisions on the attending physicians’ legal liability, or even in ethical terms, their role regarding fulfillment of medical duty. We will explore these questions in conventional medical practice, and in clinical research, with reference to the basic instruments of the common European legislation that also determines the general framework for specific national regulation in the European countries.

## 2. Data Collection and Processing

Over the last decades, progress in biomedicine has been closely associated with health data management thanks to continuously enhanced technological abilities that we dispose for data collection and processing. Current applications based on extensive personal data use that include e-prescription systems and e-health records characterize the regular performance of diagnostic, preventive, and therapeutic medical acts. These applications indicate the importance of AI components in data handling, providing immediate and accurate responses to the physician’s input [3] (pp. 33–37).

With the progress of Medical Genetics and the opening of a new era towards Personalized Medicine [2] (p. 409), [4] (pp. 21 et seq, 41 et seq), the role of data collection becomes crucial. As the current expression of that new era, Precision Medicine intends to develop tailor-made health services and therapeutic means pertinent to specific profiles of patient groups or even individual patients [5,6,7]. In this regard, AI applications will ensure feasibility since the need for accurate and rapid data processing at this scale is obvious and cannot be met by conventional human-guided methods.

In this data-centered context, the role of ethical and legal norms is pivotal. Personal data nowadays represent a distinct value in modern societies, particularly when the subject’s identity is known or may be revealed. This is because, following the fundamental principles of a democratic society, every person enjoys a space of self-determination, which also includes elements characterizing the personality’s very essence. Thus, all information related to elements of the person’s identification and privacy, forming the distinct space of “informational self-determination” [8] (pp. 398 et seq) must remain protected from any intervention of the state or thirds if unauthorized by the person concerned. Simple data of identification such as the name, the address, the phone number, the social security number, etc., belongs to this protected space.

Most importantly, special data categories are referring to the person’s intimate thoughts or private activity. Here, protection is stricter in legal terms, as any unauthorized disclosure of these data to thirds may severely damage the data subject in various social situations. In this category of “sensitive” data belong the person’s biological characteristics (genetic, etc.), personal health information, as well as philosophical or political or religious beliefs, information on friendly or sexual relationships, etc. (GDPR, art. 9).

Informational autonomy illustrates the ethical ground for personal data protection in general terms, justifying specific legal measures in relevance. Nowadays, all democratic countries have adopted laws that govern this area with detailed provisions setting up specific control mechanisms for preventing or sanctioning violations. In Europe, “guardians” of the system are the data protection authorities (GDPR, Chapter VI) enjoying an independent administrative status and the courts. The European legal framework is embedded in the General Data Protection Regulation (GDPR), an instrument that binds all EU member states and also governs data transfer and handling in non-EU countries (GDPR, Chapter V). This means that a non-EU country (including countries outside Europe, such as the USA, Canada, China, Australia, etc.) needs to demonstrate compliance with the standards of the GDPR for receiving and processing data deriving from the EU (GDPR, art. 44). Therefore, the legal relevance of the GDPR is broader in geographical terms, which makes it really influential when the issue is to promote health data collection and processing at a global scale.

On the other hand, the GDPR does not regulate data protection exhaustively. As a product of states’ negotiations and compromising, it leaves considerable discretion of regulation to national laws in EU member states, which sometimes leads to diverse modes of implementation in each separate national legal system. Thus, even if the form of an EU’s “Regulation” represents binding legislation directly enforced in the member-states’ legal systems, in the example of GDPR, national decision-making in relevance continues to hold a substantial normative role (GDPR, art. 9 (para 3, 4), 23, 46, 49, etc.).

Given this regulatory context in Europe, it is essential to distinguish categories of information relevant to health that may be accumulated at a scale of databases promising the formation of AI algorithms suitable for supporting clinical decisions. This is because the data protection regulation focuses on identifiable (GDPR, art. 4 (1)) health data exclusively, not on mere statistical data, which remain anonymous. Statistical data deriving from epidemiological studies cover a significant part of the databases’ content, and their processing is vital for the formation of AI algorithms. Still, that information remains indifferent for the law, since, in principle, there is no possibility of detecting the data subjects. 

Under the GDPR’s regime, data may be identifiable in to two categories: either data with known subjects, when the identity matters in processing given its purposes, or “pseudonymized” data, after codification, when the identity is in principle irrelevant to the processing purposes, but still may be detected if the code of anonymization is accessible [9] (pp. 663–664). Thus, personal data are equated only to identifiable data in the strict legal terminology, and data protection refers to the above two categories exclusively. 

The collection of personal data raises specific issues. Data sources are either an original collection based on direct contact with the data subjects (healthy persons or patients) or an already existing database, available for further use with different purposes than the original ones. For example, databases in hospitals comprising medical health records of patients or in insurance facilities or workplaces may be of interest for further use, particularly research use. 

It is evident that the existing databases of health data are of crucial importance for forming collections on the scale of big data in order to achieve a statistically valid multifactorial volume for testing AI applications. On the other hand, new data collections from a particular group of persons ensure that new research topics will be addressed that data already stored for other purposes cannot cover. Thus, a big comprehensive database promising the design of AI algorithms suitable for medical decision-making needs to exploit the massive material of existing databases, and also run new research to accumulate information responding to new questions of clinical importance [1] (p. 2).

Bearing in mind the above classification, the GDPR establishes a critical differentiation in data protection, focusing mostly on the issue of the subject’s informed consent as a prerequisite for ensuring personal control over any possible data use (GDPR, art. 7, 8, 9 (a)).

First, for new collections of health-related data, the subject’s informed consent is always necessary since new research objectives involve direct contact with investigators asking for such data. Still, in contrast to the previous regime governing data protection in the EU (Directive 95/46) the GDPR does not consider specific consent as a strict requirement. Following the recital 33 of its explanatory part, a consent of “generic” nature may be sufficient if determining a broader framework for the data’s secondary use [10] (p. 660), namely for future research purposes, on the evident condition that this does not mean a general permission of any research use, a “*carte blanche*” granted to investigators. Furthermore, the option that is given by recital 33 always presupposes that this “generic” consent fulfills the conditions of freedom, which is not the case when there is a “clear imbalance between the data subject and the controller” (meaning an unequal position of them, due to relationships of dependence, etc., according to recital 43 of the explanatory part). On the other hand, the original consent is required even if investigators apply data pseudonymization, and processing excludes the possibility of access to subjects’ identities, as far as the link between data and identities exists and may be known to one or more persons of the research team.

Second, regarding collections of already stored data, initially gathered for other reasons (clinical or not), the law allows further processing for new research purposes, even without the data subjects’ fresh consent, on the condition that the purposes of the secondary use of data are compatible or relevant to the research purpose that justified the original collection of data, following art. 6 para 4 (a) of the GDPR. In that case, informed consent is not considered a necessary mechanism of control for data protection. This provision facilitates research activities significantly, since to repeat communication with subjects that consented initially to the data use in other settings is practically impossible. Nevertheless, the GDPR does not leave the data protection uncontrolled. Substitute safeguards need to be in place necessarily, after specific national legislative measures (GDPR, art. 9 para 2 j) which indicatively may require pseudonymization or other technical methods to ensure confidential processing (installing firewalls, etc.). Most EU member states have already enacted such specific legislation for the implementation of the GDPR’s provision.

Third, when it comes to the use of anonymous (or “anonymized”) data, that is, data with unknown or untraceable identity, any research use is allowed with no engagement of data protection control mechanisms. Indeed, this information is not conceptualized as “personal data” by the law, similarly to what happens with statistical data. This is also important, as it covers collections of stored data that may be transferred to research facilities after anonymization at their source, with no involvement of the research team whatsoever. For example, suppose a big data collection supporting Precision Medicine’s objectives that includes partial collections of health data from private insurance companies: if these companies have performed the data anonymization in situ before transferring the collections to the research facilities, no data protection issue occurs, as researchers of the latter have no access to the anonymization procedure.

Nevertheless, in the context of big data, we must admit that neither pseudonymization nor even anonymization at the source of data secures protection of the data subjects. Indeed, the massive amount of multifactorial information from multiple sources makes possible the development of specific algorithms that may lead to findings detecting the subjects’ identity even in these data categories, following a methodology of “deep mining” analysis in which the role of AI is of course critical [10] (p. 661). This fact challenges the efficiency of the legal provisions mentioned above, which means that, eventually, the most reliable preventive mechanism for data protection remains the subjects’ informed consent prerequisite. At least for new data collections, the generic model of informed consent, as acknowledged by the GDPR, could ensure the development of big databases without compromising the data safety or the research potential. 

As a last means of protection, the GDPR recognizes specific rights for the data subjects (GDPR, Chapter III) that are fully enforceable before the courts, representing the “coercive” dimension of data protection in case of violation. Any unauthorized identification of anonymized data through “deep mining” analysis in the context of big data or other methods is, therefore, subject to administrative or judicial control under the light of these specific rights. Amongst them, the “right to erasure,” namely the subject’s legal option to ask for complete removal of his/her personal data from a database, is the most crucial here (GDPR, art. 17). Although the GDPR mentions significant exceptions regarding this right exercise, based on public interest reasons (such as public health or safety reasons), the data controller is in principle fully responsible for complying when the data subject files a relevant application. Exceptions can be considered only if their reason is specifically justified and documented. In this strict context, a potential identification of originally anonymized data in a big database would be legally unacceptable if not associated directly with evident priorities in public health. 

## 3. Artificial Intelligence and the Medical Duty

Besides the issues related to data handling, questions concerning AI applications in medical decision-making should also be considered. To what extent can a physician’s decision regarding a specific patient rely on automatically yielded guidance after AI data processing?

This is a problem relevant to the “interpretability” of AI systems [11] (pp. 3 et seq), particularly when self-learning (unsupervised) systems are engaged in conventional clinical practice or clinical research. Indeed, self-learning systems cannot be addressed as conventional medical instruments that support the physicians’ practice so far, or even as supervised AI systems, where close dependence on the human initiative (and responsibility) of the system’s expected outcomes still exists [2] (p. 399), [12] (p. 419). Self-learning ability means a certain degree of machines’ self-programming after evaluating massive information deriving from a big data biobank, which includes the relevance of existing data to specific clinical contexts. Compared to conventional medical instruments, here we have a process of data appraisal beyond direct human control. Although the algorithms that are developed for such systems are evidence-based, their automatic outcomes for supporting medical decision-making escape by default from the area of knowledge not only of ordinary medical practitioners but also of these intelligent systems’ developers raising questions on transparency regarding their functional characteristics [4] (p. 27), [13] (pp. 6 et seq). An appropriate legal methodology for addressing cases of AI developers’ professional liability should be adopted here [14] (pp. 393 et seq).

There is a clear ethical question here on whether the use of such systems meets the principles of the essential medical duty in the patient/doctor relationship, that is, the “beneficence, non-maleficence” and the informed consent (as expression of personal autonomy) requirements [15] (pp. 118 et seq, 155 et seq, 217 et seq). From a legal point of view, this question also refers to the extent of medical liability, especially when medical malpractice occurs. In any sense, accountability is a general problem related to the use of AI systems that influences professional liability, and not only in Medicine [4] (pp. 29–30, 236 et seq). Is it possible, then, to accuse a physician for medical malpractice based upon guidance from unsupervised self-learning AI systems that resulted in the patient’s harm?

To answer this question, we need, first, to highlight some elements of conventional clinical practice. Conventional medical acts should be based on two conditions: (a) the physician’s performance *lege artis*, namely according to the standard of care [16] (p. 6), which also includes compliance with relevant protocols, and (b) the patient’s informed consent (or choice), which involves patients in decision-making. On the one hand, these two prerequisites reflect the two ethical principles already mentioned; on the other, they determine the framework within which the liability of doctors should be judged in concrete legal terms. The current legislation in Europe (at the level of international instruments and mostly of national laws) contains specific provisions referring to these prerequisites, making their content legally binding. 

(a) Medical performance *lege artis* means that we admit as an axiom the existence of objective scientific norms in the framework of which doctors may exercise their activity. This does not contravene the doctors’ scientific independence and freedom of thought; it only excludes absurd practices with no scientific evidence, contrary to “professional standards” (art. 4 of the Oviedo Convention). 

There is no doubt that often the evidence issue is vague, as diverging scientific opinions cannot be excluded; therefore, opinions expressing minorities in the scientific community cannot be considered by definition absurd. Still, the axiom requires that, at least, we need firm scientific justification for accepting a certain medical art as compliant to the *leges artis.* The era of evidence-based medicine contributed to the clarification of these problems. Protocols containing specific and detailed guidelines are now developed based on the substantial progress of clinical research and the statistical reliability of research findings. These normative instruments significantly facilitate doctors’ good practice, and prevent the occurrence of severe malpractice incidents. Moreover, in the context of evidence-based medicine, the role of data processing, statistics, and mathematics became crucial several decades before the emergence of AI systems. This is a crucial point in our approach.

(b) On the other hand, the patients’ informed consent or choice holds a key role in medical practice (Oviedo Convention, art. 5, 6), even if the physician’s performance relies on machine support and guidance based on complex calculations with the use of advanced technology. This means that the physician always needs to provide appropriate information to the patient and obtain relevant consent before acting. The use of AI systems and their expected benefit or risks following evidence-based criteria definitely belongs to the content of information and possibly influences the consent, particularly of an expert patient. Nevertheless, there is no reason to exclude from this general rule even self-learning AI systems, on the condition that the patient has consented to this involvement of a “substitute” medical knowledge, and been assured that the final decision for the medical act in relevance lies on the attending physician’s direct control. Under the medical liability’s point of view, this latter element is the only decisive. Indeed, supposing that the attending physician has minimal or no specific technical expertise about the precise details of a medical instrument’s structure and function (which is the usual case), it is sufficient to demonstrate awareness of potential benefits and risks from its use to fulfill the law’s conditions on liability. The quality of patient information is the legal guarantee for this. 

Certainly, AI unsupervised, self-learning applications are characterized by an “opaque” element that remains uncontrolled by human users [4] (p. 27), [11] (p. 15), [12] (pp. 420,421). Still, in the end, what matters is the final decision of physicians about the medical act in relevance. Physicians should take the risk even for this uncontrolled element of AI systems if they believe that the benefits are more important than the potential negative implications from the use of these systems in a particular case. It is worth noting here that, in terms of medical liability, what we expect from physicians is a *lege artis* performance only, even if the final result could be non-beneficial for the patient. Medical liability concerns only criteria of good practice; therefore, if the use of AI self-learning systems is evidence-based in similar cases, no differences exist comparing to the use of conventional medical devices. 

Yet, this is true for conventional medical practice when evidence-based rules are in place. Can we suggest the same for practice in clinical research [2] (pp. 409 et seq, 413 et seq)? How appropriate is the experimentation in clinical trials with AI self-learning systems when no evidence-based criteria exist, and the question is precisely to identify such criteria? Under the medical liability view, this is a difficult problem to the extent that the “opaque” element of AI remains uncontrolled even by experts as mentioned above. An ethical question arises as well: Are we allowed to involve volunteers in experimental procedures when part of these remains beyond the investigators’ direct control? 

Again, the informed consent prerequisite is the only guarantee, here, if we assume that the information provided to the volunteers clearly includes the involvement of self-learning AI applications in the clinical trial’s development, mentioning potential risks and specific measures to be taken for preventing them. The difference that may arise compared to conventional clinical acts is that, in clinical research, we have to cope with a great deal of uncertainty by definition; therefore, the degree of risks may be unacceptable with the use of such systems. 

Nevertheless, risk acceptance is still a matter of the volunteer’s free decision. Certainly, in clinical trials, the informed consent prerequisite has limited impact compared to conventional medical acts since the law requires previous approval of the research protocol’s scientific and ethical appropriateness. This means that volunteers are invited to consent only on the condition that minimal evidence on safety and risk/benefit assessment has been obtained and confirmed by the approval mentioned above (Oviedo Convention, art. 16, 17, Directive 2001/20, art. 3, Regulation 536/2014, art. 28). In our example, minimal evidence needs to refer to the AI algorithms’ specific characteristics and self-learning operation as manifested in previous pre-clinical tests. As it happens with the new molecules’ in vivo testing in pre-clinical studies demonstrating the expected influence on animal organisms, this step seems both necessary and sufficient to ensure the minimal evidence that allows the protocol’s ethical approval; moreover, it justifies seeking the volunteer’s informed consent. This analogy is defensible because, in terms of safety, the impact of an experimental substance on a living organism’s vital functions is not less risky than the AI’s self-learning guidance of clinical decision-making since both are based on a rational assessment of scientific data. In this comparison, the expected guarantees for a positive outcome have the same degree of reliability. In other words, the degree of uncertainty is comparable, particularly if no option of return to the condition before the intervention is ensured. 

## 4. Conclusions

The novel element that AI applications bring to biomedicine is the mobilization of machine-controlled inputs in decision-making regarding either conventional or experimental medical acts. Relying on AI systems’ self-learning operations, this technological development facilitates immensely the accurate appraisal of data relevant to specific clinical situations based on robust medical evidence. There is no doubt that machine self-learning guarantees what the human medical practice, even of highly experienced experts, cannot provide, namely to yield practical guidance timely from a work of massive data processing. Yet, the cost that we need to accept for that is not trivial.

First, there is a cost regarding the need to handle big health data, which refers to risks occurring for the protection of identifiable personal data. The latter’s collection and processing involve procedures that, in principle, may guarantee protection, but still the risk is persistent at that scale, since nothing is “automatically” in place, and specific responsibilities of many people acting in that field need to be considered. 

In Europe, following the GDPR’s regulation, data controllers, data processors, and data protection officers are the main responsible persons, here (GDPR, art. 24, 28, 37) Since the development of more advanced AI systems is embedded in the permanent accumulation of massive information, a particular problem of data transfer emerges, given that no unified binding legislation and controls exist at the global scale, and additional procedures for ensuring the data subjects’ rights need to be observed.

Second, a further cost is related to the medical performance as such, that is, the extent of medical acts’ dependence on machine-controlled guidance. Inevitably, in unsupervised, self-learning AI medical applications, we have to cope with an “opaque” element that escapes the attending physician’s direct control, although it still affects medical liability. 

We argued that in conventional clinical practice, the attending physician remains responsible (a) for using such systems only if they are evidence-based, and (b) for providing appropriate information to the patient that includes necessarily a risk/benefit appraisal for these systems; if the patient consents on the basis of that information, the essential legal requirements for assuming good medical practice are fulfilled. 

In the clinical research context, evidence on the use of experimental self-learning AI applications is under investigation by definition; therefore, the above model needs to be reconsidered since we have to deal with an essential element of uncertainty that may entail risks for the volunteers’ health. Here, we propose an analogy between the AI application’s uncertainty and the uncertainty deriving from the use of experimental molecules in interventional clinical studies. We may assume that the levels of potential risks are similar. Guarantees for both acts’ suitability remain, on the one hand, the successful results that pre-clinical trials demonstrate and, on the other, the information provided to the clinical trial’s volunteers for the use of such experimental methods. 

Under this view, we can conclude that legally speaking at least, there is already a rationale framework for appraising the issue of medical liability, even when the use of self-learning AI systems cannot be equated to that of conventional medical instruments, where usually no “opaque” characteristics escaping from the physician’s direct control exist.

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
