# Peer review of "Artificial Intelligence in Biomedicine: A Legal Insight"

_biotech, 2021, doi:10.3390/biotech10030015_

Round 1
Reviewer 1 Report
The author makes some general statements that are not entirely incorrect but risks confusion or misunderstanding by readers if not explained with greater clarity or reference to other parts of the GDPR
For example, the author states "the law allows further processing [of previously collected data] for new research purposes, even without the data subjects' fresh consent". This is not entirely true, as the GDPR requires that the new purpose be compatible with the original purpose. Another example relates to the issue of consent. The author writes "the GDPR does not consider specific consent as a strict requirement" but recital 33 needs to be understood in the broader context of other parts of the GDPR that relate to consent (ie. recital 43 and article 6 to begin with). These are only two examples where readers could benefit from greater clarity. There are several other aspects throughout section 2 of the article that requires greater attention and care.
Section 3 could benefit from some reference to the literature on black box medicine and transparency when discussing and framing the issue of ethics and liability.
Author Response
Dear Reviewer
Thank you for your valuable input. Below you can see replies to your comments:
“The author makes some general statements that are not entirely incorrect but risks confusion or misunderstanding by readers if not explained with greater clarity or reference to other parts of the GDPR. For example, the author states "the law allows further processing [of previously collected data] for new research purposes, even without the data subjects' fresh consent". This is not entirely true, as the GDPR requires that the new purpose be compatible with the original purpose. Another example relates to the issue of consent. The author writes "the GDPR does not consider specific consent as a strict requirement" but recital 33 needs to be understood in the broader context of other parts of the GDPR that relate to consent (ie. recital 43 and article 6 to begin with). These are only two examples where readers could benefit from greater clarity. There are several other aspects throughout section 2 of the article that requires greater attention and care.”
- Clarifications in relevance are provided in the revised text (lines 138 – 142, 148 – 150). I agree that these particular points could raise misunderstandings.
“Section 3 could benefit from some reference to the literature on black box medicine and transparency when discussing and framing the issue of ethics and liability”.
- More references to additional literature regarding transparency, interpretability, opacity, and accountability /liability have been included in Sec. 3.
Please, see the attachment

Reviewer 2 Report
Very well written paper.
The paper includes a critical analysis of data protection legislation on artificial intelligence and examines how it can regulate it for the benefit of patient protection. It explores mechanisms of the GDPR that can balance the benefits of using artificial intelligence with the protection of the individual. Furthermore, the paper explores the ethical question whether the use of such systems meets the principles of the essential medical duty in the patient/doctor relationship, that is, the “beneficence, non-maleficence” and the informed consent (as expression of personal autonomy) requirements. The analysis is persuasive, written in understandable language, ends up to an understandable conclusion and leaves no gaps to the reader.
Author Response
Thank you very much for your positive comments, and really encouraging review!